# Perceptions and Experiences of the University of Nottingham Pilot SARS-CoV-2 Asymptomatic Testing Service: A Mixed-Methods Study

**DOI:** 10.3390/ijerph18010188

**Published:** 2020-12-29

**Authors:** Holly Blake, Jessica Corner, Cecilia Cirelli, Juliet Hassard, Lydia Briggs, Janet M. Daly, Malcolm Bennett, Joseph G. Chappell, Lucy Fairclough, C. Patrick McClure, Alexander Tarr, Patrick Tighe, Alex Favier, William Irving, Jonathan Ball

**Affiliations:** 1School of Health Sciences, University of Nottingham, Nottingham NG7 2HA, UK; llxlb15@nottingham.ac.uk; 2NIHR Nottingham Biomedical Research Centre, Nottingham NG7 2UH/NG7 2RD, UK; mbzjgc@nottingham.ac.uk (J.G.C.); mrzpcm@nottingham.ac.uk (C.P.M.); alex.tarr@nottingham.ac.uk (A.T.); paddy.tighe@nottingham.ac.uk (P.T.); will.irving@nottingham.ac.uk (W.I.); jonathan.ball@nottingham.ac.uk (J.B.); 3University Executive Board, University of Nottingham, Nottingham NG7 2RD, UK; jessica.corner@nottingham.ac.uk; 4School of Medicine, University of Nottingham, Nottingham NG7 2UH/NG7 2TU, UK; mzycc9@nottingham.ac.uk (C.C.); juliet.hassard@nottingham.ac.uk (J.H.); 5School of Veterinary Medicine and Science, University of Nottingham, Loughborough LE12 5RD, UK; janet.daly@nottingham.ac.uk (J.M.D.); M.Bennett@nottingham.ac.uk (M.B.); 6School of Life Sciences, University of Nottingham, Nottingham NG7 2RD/NG7 2UH, UK; lucy.fairclough@nottingham.ac.uk; 7Faculty of Registrars, University of Nottingham, Nottingham NG7 2RD, UK; alex.favier@nottingham.ac.uk

**Keywords:** COVID-19, SARS-CoV-2, coronavirus, virus, disease outbreaks, young people, students, health promotion

## Abstract

We aimed to explore student and staff perceptions and experiences of a pilot SARS-CoV-2 asymptomatic testing service (P-ATS) in a UK university campus setting. This was a mixed-method study comprised of an online survey, and thematic analysis of qualitative data from interviews and focus groups conducted at the mid-point and end of the 12-week P-ATS programme. Ninety-nine students (84.8% female, 70% first year; 93.9% P-ATS participants) completed an online survey, 41 individuals attended interviews or focus groups, including 31 students (21 first year; 10 final year) and 10 staff. All types of testing and logistics were highly acceptable (*virus*: swab, saliva; *antibody*: finger prick) and 94.9% would participate again. Reported adherence to weekly virus testing was high (92.4% completed ≥6 tests; 70.8% submitted all 10 swabs; 89.2% completed ≥1 saliva sample) and 76.9% submitted ≥3 blood samples. Students tested to “keep campus safe”, “contribute to national efforts to control COVID-19”, and “protect others”. In total, 31.3% had high anxiety as measured by the Generalized Anxiety Disorder scale (GAD-7) (27.1% of first year). Students with lower levels of anxiety and greater satisfaction with university communications around P-ATS were more likely to adhere to virus and antibody tests. Increased adherence to testing was associated with higher perceived risk of COVID-19 to self and others. Qualitative findings revealed 5 themes and 13 sub-themes: “emotional responses to COVID-19”, “university life during COVID-19”, “influences on testing participation”, “testing physical and logistical factors” and “testing effects on mental wellbeing”. Asymptomatic COVID-19 testing (SARS-CoV-2 virus/antibodies) is highly acceptable to students and staff in a university campus setting. Clear communications and strategies to reduce anxiety are likely to be important for testing uptake and adherence. Strategies are needed to facilitate social connections and mitigate the mental health impacts of COVID-19 and self-isolation.

## 1. Introduction

The coronavirus disease (COVID-19) pandemic caused by the virus SARS-CoV-2, resulted in a United Kingdom (UK) national lockdown in March 2020 and stay-at-home orders, followed by long-lasting national social distancing measures and travel restrictions (gov.uk/coronavirus). Throughout this time, universities have remained open in the UK. The University of Nottingham hosts the first veterinary school in the UK to operate a dual intake system, which began at the start of the academic year in September 2019. The first cohort of 2020 (Cohort 1) were disrupted by the national lockdown, meaning that teaching started remotely in the weeks prior to the initiation of small-group face-to-face teaching, which started in July [1]. The potential for COVID-19 transmission on university campuses is high [2]. In preparation, the university implemented health and safety measures across teaching buildings and accommodation to allow for the safe return of students onto a single, semi-rural campus. Since university-age students have a higher prevalence of SARS-CoV-2 infection [3], and higher rates of asymptomatic infection [4], this included a 12-week mass COVID-19 testing service piloted as a health protection approach to the early identification and control of potential outbreaks on campus. While the institution has certainly capitalised on digital innovation in remote learning [5], the intention of this mass testing approach was to enable face-to-face teaching that cannot be delivered remotely, such as essential animal handling and health and safety skills that would be required prior to embarking on work experience placements, a core element of training. The new students in Cohort 1, therefore, joined the University of Nottingham at the height of the UK lockdown, and together with existing final-year students undertaking clinical rotations, were among the first students in the UK to experience SARS-CoV-2 asymptomatic testing, social distancing and hygiene measures in a university setting.

To date, there is only one prior published study assessing the feasibility or acceptability of a universal programme for SARS-CoV-2 testing on a UK university campus [6] although this reports the findings of a shorter pilot programme (2 weeks), using only PCR (polymerase chain reaction) swab tests and did not assess the acceptability of saliva tests or antibody tests or measures of anxiety or any other aspects of student wellbeing. Berger Gillam and colleagues primarily focused on costs, guidance materials, logistics, laboratory and data processes and a user-facing web application, and participant acceptability was determined only from email communications and an 11-item survey [6].

To our knowledge, this is the first study to explore student and staff views towards university-based asymptomatic testing programmes alongside perceptions of COVID-19 risk, anxiety of student participants, reasons for uptake and the facilitators and barriers to testing adherence. The aim of this study was to (i) evaluate the acceptability of a pilot COVID-19 asymptomatic testing service to students and staff on a university campus; (ii) describe benefits and barriers to programme engagement and testing adherence; (iii) establish whether there are any relationships between adherence to testing, and students’ anxiety levels, COVID-19 risk perceptions, views towards protective behaviours (social distancing, self-isolation); (iv) identify any perceived broader impacts of participation in an asymptomatic testing programme for students or staff to assist with recommendations for future testing services in higher education settings.

## 2. Methods

This was a mixed methods study to evaluate the deployment of a pilot COVID-19 weekly asymptomatic testing service (P-ATS: Figure 1) offered to students and staff who had face-to-face teaching responsibilities during this period. The study explored the uptake, adherence, acceptability and experiences of the P-ATS in students as well as assessing students’ anxiety and risk perceptions and the perceptions of university staff towards P-ATS implementation. 

The study comprised of (i) a structured online survey for students administered at the end of deployment, (ii) a qualitative study involving interviews and focus groups with university students and staff conducted at the programme mid-point (students) and at the end of deployment (students and staff). The study design adheres to the standards for reporting qualitative research [7] and the consolidated criteria for reporting qualitative research guidelines [8] (see Appendix A). A total pool of 215 undergraduate students (150 in their first year and 65 in their final year) and 70 staff were eligible to participate in the P-ATS. 

### 2.1. Pilot Asymptomatic Testing Service (P-ATS)

The P-ATS was a pilot programme for SARS2 surveillance conducted in the early phases of the SARS2 pandemic at a semirural campus of a university in the UK. The objective of the programme was to define the baseline SARS-Coronavirus type 2 (SARS-CoV-2) infection rate and seroprevalence in a cohort of university students and staff and to measure changes over time.

Participation in P-ATS was on a voluntary basis and aimed to be complementary to the national testing strategy in the United Kingdom (UK). P-ATS was a completely independent initiative within the University to explore the value of SARS-Cov-2 testing in asymptomatic students. National guidelines for actions to be taken by symptomatic students were followed throughout. Any asymptomatic students testing positive in this pilot were advised to undertake a “Pillar 2” test, the results of which would be paramount (the University laboratory testing was a research test only), and hence would feed into the UK National Health Service (NHS) Test and Trace [9] mechanisms, if positive.

The P-ATS was offered to all 150 students in their first year (new arrivals in 2020 Cohort 1) as well as 65 final-year students going out into practice rotations in the Autumn term. A select group of 70 university staff who had face-to-face contact with students during the study period were offered the opportunity to join P-ATS if they wished, at any point during the programme. 

The P-ATS was primarily targeted to students in their first year of study who were living in university accommodation on campus. All first-year students who had arrived in 2020 Cohort 1 were therefore eligible for the P-ATS and were offered the full programme which included a total of 12 PCR tests to be completed weekly over 12 weeks (10 swab tests, and 2 saliva tests) and up to 6 antibody tests from July to October 2020. In addition, a number of students in their final year were invited to join the P-ATS in September 2020. Eligible final-year students were those who were due to start certain 2-week rotation placements that required them to undertake PCR testing prior to attendance. A select number of final year participants therefore participated in the P-ATS between September and October 2020. 

Newly arriving first-year students from 2020 Cohort 1 were accommodated on campus in cluster flats treated as a “household” and attended teaching sessions in “bubbles” through the study period to avoid exposure to larger groups of people. Final-year students mostly lived off campus and attended clinical rotation placements in the community during this year of study. The P-ATS start for eligible final-year students was staggered since they joined at different times according to academic timetabling and the start date of relevant placements. Final-year students joined the programme at different times, had fewer tests offered in total, with some students taking only one test, and others taking more if they changed rotation placement during the study period. Therefore, the total number of tests offered varied between students, although the testing offers and start date were consistent for those who were in their first year. The first- and final-year students who accessed P-ATS, therefore, had a different experience of both the service (with only first years offered the complete programme), and of university life more broadly.

For all P-ATS participants, swab, saliva and antibody test kits were collected by P-ATS participants, tests were self-administered and participants deposited samples at dedicated collection points on the university campus with social distancing rules applied.

The complete programme consisted of:(a)SARS2-PCR tests offered weekly for the 12-week study period (10× swab and 2× saliva);(b)SARS2-antibody tests offered alternate weeks (6 x self-sampled finger-prick dried blood sample).

Individuals testing negative were informed by email correspondence to their cohort (indicating that all positive cases had been contacted). Individuals testing positive were personally advised of their result by a clinical virologist, and a central university team was notified so that the student could be safely cared for. This process included notification of Public Health England so that official test, track and trace could take place [9]. 

### 2.2. Participants, Recruitment and Sampling

All students who were eligible to take part in the P-ATS were invited to fill out the post-P-ATS evaluation survey (*n* = 215). Participants for the mid-point and post-P-ATS interviews and focus groups were recruited from the total pool of 285 individuals (first-year students: *n* = 150; final-year students: *n* = 65; staff: *n* = 70) who had been invited to take part in P-ATS. All students were invited to complete the survey, whether or not they had taken part in P-ATS. All students and staff who had taken part in P-ATS were invited to attend an interview or focus group. Recruitment to the mid-deployment group interview with students took place in weeks 6 and 9. Recruitment to the post-deployment survey and student interviews and focus group commenced in mid-October 2020 and continued for 16 days through weeks 12–14. Recruitment to the staff focus group took place post-deployment in week 13. The staff focus group included participants in clinical or non-clinical teaching or research roles, senior leadership, support staff (e.g., technicians) and staff with pastoral or welfare roles (e.g., tutors, student experience administrators). Students who were invited to interviews and focus groups included individuals of any gender, those living on or off campus, and those in their first or final year of study.

Ninety-nine students returned the post-P-ATS evaluation survey (46% response rate), 52 students and staff consented for the qualitative study via an online form and 41 subsequently took part in interviews or focus groups during the 16-day data collection window. Table 1 shows basic demographic information for interview participants.

### 2.3. Online Survey

All students who were eligible to take part in the P-ATS were invited to fill out the post-P-ATS evaluation survey using Jisc Online Surveys (see Appendix A). The survey contained a mixture of closed and open-ended free-text questions exploring participants’ reasons for participating in the P-ATS (or not), experiences and engagement with the testing, perceived barriers and benefits of the programme and suggestions for improvement. Items explored students’ experiences of self-isolation and social distancing, COVID-19 risk perceptions and anxiety measured by the Generalized Anxiety Disorder scale (GAD-7) [10]. Generalized anxiety disorder (GAD) is one of the most common mental disorders, and the GAD-7 has demonstrated strong psychometric properties in population-based samples [10,11]. The measure has a range of 0 to 21, and a score of 10 or greater on the GAD-7 represents a reasonable cut point for identifying cases of GAD, with high sensitivity (89%) and specificity (82%) [10].

### 2.4. Qualitative Interviews and Focus Groups

The study explored the perceptions and experiences of staff and students who were invited to take part in the P-ATS. Qualitative data were collected from interviews and focus groups as well as free-text questions from the online survey. Thirty-one student participants (21 first year, 10 final year) took part in six individual interviews (*n* = 6), eight small group interviews with 2–3 participants (*n* = 20), and one focus group with 4–5 participants (*n* = 5) all held online using video-conferencing facilities. Two student group interviews took place at the programme mid-point (in weeks 6 and 9, respectively), all other interviews took place at the end of the P-ATS (weeks 12–14). A single focus group was held with 10 staff participants after programme end, at week 13. Interviews and focus groups were facilitated/moderated by a psychologist experienced in running focus groups (HB), and a study researcher who was a medical trainee (CC). Both had undertaken training in qualitative research and interview skills. Focus groups were conducted according to recommendations from NHS England’s focus group guide [12]. All interviews and focus groups followed the same questioning route (see Appendix A), were audio-recorded and transcribed verbatim.

### 2.5. Reflexivity Statement

The research team members reflected on the impact of their background, training, beliefs and relationship to the research topic. Eleven of the authors conceptualised or were involved in operationalising the P-ATS being evaluated (JB, WI, JC, MB, JD, LF, PM, AF, JCh, AT, PT), although these researchers were not involved in evaluation data collection or analysis. Survey data were analysed by a researcher who was not involved in recruitment, intervention delivery or data collection (JH). Of the researchers who collected qualitative data, one had medical training (CC), and one was a psychologist (HB). Of the researchers who conducted thematic analysis, one was a nurse who had not collected data (LB), the other had moderated focus groups and conducted interviews (CC) which may have influenced interpretation but was mitigated by team reflexivity.

### 2.6. Data Analysis

Survey data were analysed using IBM PASW SPSS Version 25.0 (IBM, Armonk, NY, USA) (Appendix A). Data cleaning procedures (e.g., identification outliers and missing data analysis) and key statistical assumptions underpinning *t*-tests, correlation, and linear regression (normality, linearity, homoscedasticity, and independence) were examined prior to data analysis. Qualitative data from interviews, focus groups and open-ended survey questions were analysed using inductive thematic analysis, which benefits from theoretical flexibility and simplicity in the identification of qualitative themes [13]. This process included the in-depth familiarisation and coding of data using NVivo 12 software, before sorting data in broader thematic concepts which represented sections of the data, later refined into the development of five key themes, and 13 subthemes. Two researchers (LB/CC) analysed qualitative data, using thematic analysis [13]. As this was an evaluation of a pilot uncontrolled complex intervention in a real-world setting, intended to directly inform ensuing mass SARS-Cov-2 testing approaches, a pragmatic and time-sensitive approach was taken to analysis. Three researchers were involved in the qualitative analysis (LB, CC, HB). One researcher coded all the interview data and generated the initial themes (LB), a second researcher (who had conducted interviews) then independently coded a subsample of four randomly selected transcripts, in order to compare and agree on themes through discussion. A third researcher (who had conducted interviews) then reviewed all the transcriptions to crosscheck against themes, confirm the themes and resolve any discrepancies between coders (HB). Consensus on the themes was achieved through discussion between all researchers. Combining qualitative data from different data sources and using two researchers for coding and analysis, enabled data and investigator triangulation.

### 2.7. Patient and Public Involvement

Student and staff views informed the study design and interview questioning guide at the point of study conception, via a Patient and Public Involvement and Engagement (PPIE) group. Students expressed a preference to be able to choose between individual or small group interview, and staff preferred to participate in a single large focus group. Study findings will be disseminated to all participants through this publication and lay summaries disseminated via the participating university.

## 3. Results

### 3.1. Survey Results

The study sample included 99 respondents (93 P-ATS participants, 6 nonparticipants). Sample characteristics are shown in Table 2. Mean age was 20.36 years (SD 1.69). Table 3 provides an overview of self-reported anxiety levels within the total sample and across key groupings. Of respondents, 31.3% had high anxiety (score of >= 10) as measured by GAD-7 (43.3% of final-year and 26.1% of first-year students). Mean anxiety levels were higher in students living in private accommodation compared with students in university halls of residence. G*Power (version 3.1.9.7) [14] was used to calculate post hoc statistical power. All independent *t*-tests are underpowered (<0.8) and, therefore, the risk of false negatives was inflated and the results (including null findings) should be interpreted with caution. 

#### 3.1.1. Reasons for Testing

Students’ top three reasons for taking part were “helping to keep campus safe”, “contributing to the national effort to control the virus”, and “being involved in COVID-19 research” (see Figure 1). Table 4 provides an overview of participants’ experiences of the P-ATS. The majority of respondents reported they would take part in a COVID testing programme in the future (94.9% of P-ATS participants, 50% of non-participants), and would recommend university asymptomatic COVID testing to others (98% of participants, and 100% of non-participants). Reasons for nonparticipation were unrelated to COVID-19 or testing (e.g., not physically present at the university during this time) and there were no observable differences in demographics between participants and nonparticipants informed by descriptive statistics. Due to the small sample size group mean differences could not be tested using inferential statistics.

#### 3.1.2. Case Identification

Only four positive SARS-CoV-2 cases were identified through the P-ATS in this sample. Three of these students reported that they were notified of their positive result within 24 h by the Clinical Virologist, and one student reported that they had been notified after 2 days. All four students were compliant in notifying the university the same day using an online reporting form. All received the official test kit and all self-isolated as advised. One of the students stayed exclusively in their own room during self-isolation, the other three self-isolated within their household but had contact with other household members.

#### 3.1.3. Acceptability and Programme Satisfaction

Test kits were collected by individuals or members of their household and almost all of the participants did not report any issues with drop off and collection procedures. More than three quarters of respondents (79.6%) indicated they were confident in the outcome of their COVID-19 test result. Indicators of acceptability are shown in Table 4 and Table 5. In general, students were highly satisfied with the information they received about the testing programme (97.5%) and how the information was communicated to them (89.2%). Respondents were satisfied with the approach to communicating positive test results, but over one-third were dissatisfied with receiving negative test results via a group email (e.g., indicating that all individuals who tested positive had been informed), rather than being told their negative result individually (the process in place during the pilot deployment).

#### 3.1.4. Testing Adherence

Adherence could be meaningfully determined for students in their first year who had participated in the P-ATS, as they were the target population and had all been offered the full P-ATS provision (testing provision for final year and staff was individualised, so highly variable). Reported adherence to testing related to COVID was relatively high among first year students. Of the first-year survey respondents who had participated in P-ATS (*n* = 65), 70.8% (*n* = 46) submitted all 10 swab tests in weeks 1–10 (full swab provision), and 93.9% (*n* = 61) submitted 5 or more swabs (at least half the swab provision). With regard to saliva samples only, 89.2% (*n* = 58) of first years completed one or more samples, and 16.9% (*n* = 11) completed two or more saliva samples. For both types of test combined, 92.3% (*n* = 60) completed 6 or more tests, and 47.7% (*n* = 31) completed all 12 tests (10× swab, 2× saliva). Reported PCR test completion is provided in Figure 2, Figure 3, Figure 4 and Figure 5 for the first year P-ATS participants (*n* = 65). The change from swab test to saliva sample was initiated at week 10 of 12, in preparation for alignment with deployment of the main university testing service. Engagement willingness may be underestimated from the number of tests completed, due to brief period of test kit stock depletion during the study period.

As would be expected, due to a longer period in the P-ATS, first-year students completed significantly more PCR tests than final-year students during the pilot: X_first year_ = 10.13, SD = 2.82, *n*= 65; X_final year_= 2.68, SD = 1.12, *n* = 28; *t* (91) = 12.51, *p* < 0.001. Figure 4 stratifies these reported frequencies across year groups. Across the whole sample (*n* = 93), reported adherence to PCR testing was significantly higher in those who had been required to self-isolate at any point during the P-ATS (X_self-isolate_= 9.03, SD= 2.98, *n* = 34; X_no self-isolate_= 7.15, SD = 4.82, *n* = 59; *t* (91) = 2.19, *p* = 0.031, X_difference_: 1.99), and those with lower levels of anxiety (X_low anxiety_ = 8.70, SD = 4.24, *n* = 64; X_high anxiety_ = 6.07, SD = 3.99, *n* = 29; *t* (91) = −2.83, *p* = 0.006). Adherence to PCR testing was also higher in those students who lived on campus (*n* = 63) compared with those who lived elsewhere (X = 10.15_on campus_, SD = 2.67, *n* = 61; X_off campus_ = 3.56, SD = 2.99, *n* = 32; *t*(91) = −10.14, *p* < 0.001), although this is not unexpected, given that almost all of the on-campus students were in their first year (*n* = 62) and were offered the full 12 weeks of P-ATS. Among the first-year students specifically (*n* = 65), adherence to PCR testing was significantly higher in those with lower levels of anxiety (X_low anxiety_ = 10.71, SD = 2.47, *n* = 48; X_high anxiety_ = 8.47, SD = 3.14, *n* = 17; *t* (63) = −2.98, *p* = 0.004).

With regards antibody tests, 76.9% (*n* = 50) of first-year students reported completing three or more of the six tests offered during the P-ATS, and 41.5% (*n* = 27) of first-year students completed all six tests. Figure 5 stratifies these reported frequencies across year groups.

#### 3.1.5. Correlates of Self-Testing

Pearson and Kendall’s Tau-b correlation coefficients were calculated to examine the association between reported test compliance, mental health and programme satisfaction for all student participants in the P-ATS (combined PCR swab and saliva samples, Table 6; antibody tests; Table 7). Increased PCR test adherence was associated with increased worry about friends and family contracting COVID-19, greater satisfaction with drop-off location, increased satisfaction with how negative tests were communicated, greater satisfaction with the information received and greater satisfaction with how information was communicated. A higher frequency of completed antibody tests were associated with greater worry about friends and family contracting COVID-19, greater satisfaction with information received and how it was communicated, and satisfaction with drop-off location.

Due to the exploratory nature of this study, a stepwise entry method was used (with the exception of covariates) to specify regression analysis. The frequency of combined reported PCR testing (swab and saliva) completion was used as the dependent variable. Gender was dummy coded, with males set as the referent group. Covariates (gender and year of study) were entered in block one (forced entry), followed by student’s mental wellbeing variables (block two, stepwise entry), and satisfaction with programme services and communication (block three, stepwise entry). See Table 8 and Table 9 for overview. Two standardised residuals (>+/−3.0) were identified as extreme scores and were removed from this analysis (*n* = 91). Increased PCR test adherence was significantly associated with being in the first year of study (as expected due to the higher number of tests available for first years compared with final years), lower levels of anxiety, increased worry about contracting COVID-19 and increased satisfaction with the way in which information was communicated (see Table 8). The final regression model accounted for 78.2% of the explained variance in the dependent variable (adjusted R^2^ = 0.769, SE = 2.06). The statistical correlates associated with the frequency of antibody tests were determined with several study variables (see Table 7). Increased frequency of antibody tests completed during the pilot was statistically significantly associated with being first year of study (again, as expected due to the higher number of tests offered to first year compared with final years), lower level of anxiety symptoms, having greater worry about friends or family contracting COVID-19, increased satisfaction with the drop-off point for completed tests and greater confidence in the outcome of the test. The final regression model explained 57.9% of the total variance (adjusted R^2^= 0.579, SE = 1.54; see Table 9).

#### 3.1.6. Correlates of Mental Health

A regression analysis was conducted to examine the correlates of mental health (specifically, anxiety measured by the GAD-7) and testing procedures or concerns regarding COVID-19. A three-step hierarchical regression was conducted using a stepwise entry method. Gender was dummy coded, with males set as the referent group. The final regression model accounted for 46.2% of the total explained variance: adjusted R^2^ = 0.431, SE = 4.48. See Table 10 for overview of regression results and block entry method. Increased anxiety among students was associated with (listed in descending order of magnitude of association strength): increased worry about contracting COVID-19, decreased satisfaction with the way in which information was communicated through the pilot and increased worry about friends and family contracting COVID-19.

#### 3.1.7. Protective Health Behaviours

Perceived risk of COVID-19 (before and after the testing pilot) and the perceived importance of protective health behaviours is shown in Table 11 for the entire sample, and across sub-groups (positive tests, participants in pilot, and those that have self-isolated). Independent *t*-tests were calculated to test group mean difference. It is important to note that all *t*-tests were underpowered, with an elevated risk of false negatives (Type II error). Therefore, null results should be interpreted with caution. Those who opted not to take part in the pilot programme reported a lower perceived risk of COVID-19 than pilot participants pre- (July) and post-pilot (October) compared with those who had taken part, although the difference only reached statistical significance for the pre-pilot rating. There were nonsignificant trends towards lower perceived importance of protective behaviours in nonparticipants, those who had received a positive test result and those who had needed to self-isolate during the intervention period. However, it is important to note that, due to limited sample size of one comparison group, these tests have limited statistical power (resulting in an inflated risk of type II error) and should be interpreted with caution.

### 3.2. Qualitative Results

Five themes (with 13 sub-themes) emerged from the analysis of the qualitative data from student and staff interviews and focus groups: “emotional responses to COVID-19”, “university life during COVID-19”, “influences on testing participation”, “testing physical and logistical factors” and “testing effects on wellbeing”. A thematic map illustrating the relationships between the key themes and subthemes is provided in Appendix A. Table 12 shows a summary of key themes and subthemes and their representative quotes. Further quotations to support Themes 1–5 are provided in Appendix A.

#### 3.2.1. Theme 1: Emotional Responses to COVID-19

(i) Negative Feelings

Participants expressed complex emotional reactions to COVID-19, including physical exhaustion and a range of psychological responses such as frustration, uncertainty, anxiety and fear. Fear was alluded to not only in the context of the risk to themselves and their families of contracting COVID-19 but also in respect to a sense of guilt they would feel should they receive a positive test result and risk spreading the virus to others. While a minority were less concerned by the virus, there was widespread acknowledgement that the pandemic had impacted significantly on mental health for both students and staff, “in the beginning when it all went into lockdown it felt really alien and it made me quite anxious” (P128, female, staff, academic). Students expressed frustration with regard to the negative media representation of university students, and they perceived mixed messages from the government had encouraged them to “eat out, to help out”, and subsequently “blamed students for going out, and spreading COVID-19”.

(ii) COVID-19 around Me

Variations observed in participants’ emotional reactions to COVID-19 (students and staff) and their level of concern about COVID-19 appeared to be related directly to their personal experiences. Anxiety and fear were much more pronounced in those reporting greater personal exposure to the negative impacts of COVID-19 than those who had no prior experience of the virus or its impacts. For example, interviewees were seemingly more anxious and concerned if they had received a positive test result, had (or knew someone who had) self-isolated during the pandemic, had an underlying health condition that increased health risk, or knew someone who had become seriously ill due to COVID-19: “one of my friends was on a ventilator for ten weeks because of… being affected by COVID, he was a nurse working with elderly people so that was quite a scary experience” (P126, female, staff, academic).

(iii) Coping with COVID-19

Participants referred to a diverse range of coping mechanisms adopted in response to the virus, which included participation in the university asymptomatic testing provision, media avoidance, distractions, and an acceptance that outcomes were beyond their control, all of which appeared to reduce their anxiety. Many students alluded to peer support both with relation to shared engagement with peers in the testing processes (e.g., within household “bubbles”) and accessing support from peers during self-isolation. It was suggested that students who live off campus may have less peer support and may find periods of self-isolation more challenging than those living in halls. A small number of students had struggled to cope during the pandemic, particularly when self-isolating, and staff indicated that further support was required for mental wellbeing, particularly for those students with existing mental health concerns. “We have had situations where students have significant mental health crises during self-isolation and that has put a lot of stress on the other students in the house where they are in an isolating situation… we need to make sure we have enough support 24 h a day, seven days a week, for these students” (P129, male, staff, academic).

#### 3.2.2. University Life during COVID-19

(i) Adaptation to a New Normal

Students indicated they would rather be present at university being tested for SARS-CoV-2 (COVID-19), than go home. Some staff reported feeling initially anxious about their return to work at the university during the pandemic, but they were keen to do so and had quickly adapted to this and the regular testing. With the emotional responses to COVID-19 in mind, participants described extensive adaptations required in order to adapt to a “new normal” in the ongoing pandemic. This included compliance with national restrictions, such as social distancing and self-isolation, but also the acknowledgement of the further challenges these restrictions presented to university life with relation to social engagement (and for first year students, making new friendships) as well as impacts on teaching and learning activities. 

Overall, students and staff reported adapting well to the changes resulting from COVID-19 (e.g., online learning and remote working). During the intervention period, the experience of testing became more normalised as households engaged in testing processes together and it became a shared habit. Periods of self-isolation were seen to be the “new normal” during the pandemic, but participants raised difficulties experienced because of this. Some were disappointed to have missed family events, others spoke about the acute impact of self-isolating on mental health and the tensions that had arisen in households as a result of the mental health impact of COVID-19. First-year students who had not yet established friendship networks were concerned about missing out on university life: “there is a fear of not making friends at uni, so, when people come here, everyone wants to make lots of friends because that is where you’re here to do at uni, so it is kind of hard having those restrictions kind of prevent that.” (P98, female, Yr1, student). 

Staff spoke of the heavy impacts of the pandemic on their workload, particularly with relation to teaching and learning activities, such as the transition to remote working, re-scheduling of assessments for students who were self-isolating and the additional work required to support partial face-to-face teaching in a COVID-safe environment. One of the more significant challenges for staff was the management of student welfare. They reported escalating mental health concerns among students that were exacerbated by periods of self-isolation and compounded in the early stages of P-ATS by practical issues for students living on and off campus that were later resolved (e.g., “teething problems” around the organisation of food delivery and laundry services). With the potential for students being required to isolate more than once, staff perceived that adaptation to the new normal of COVID-19 and the success of mass testing approaches would require a substantial focus on student welfare and support for mental wellbeing.

(ii) Improving University Life

Participants made various suggestions for how the challenges of COVID-19 and adapting to the new normal might be addressed. There was a strong appetite for wider-spread testing across all campuses to maximise perceptions of safety in the student and staff body. There was a desire for more university-led events to entertain students, facilitate social activity to assist with friendship building and to reduce the boredom of self-isolation. There was a general consensus that personal contact and support is essential to reduce the fear and anxiety experienced during this time. Students requested personal support from university staff with managing the challenges of missed opportunities or learning experiences due to self-isolation (e.g., lectures or practice rotations). They spoke of the importance of regular check-ins from staff during periods of self-isolation to minimise the mental health impacts: “it doesn’t need to be a lot, simple email, not even every day like every other day, how are you doing? Do you need anything?” (P116, female, Yr1, student). Staff were commended by students for the level of support they had provided during this time. Some participants believed that there should be increased education around repercussions or regulation defiance for the minority of students that were not adhering to social distancing advice. Students acknowledged that some positive support was already in place from the university with practical tasks, including shopping, and were aware of the workload pressures that this added for staff. However, staff disclosed pressures associated with increased student support, particularly with relation to the workload and challenges associated with supporting students after they received a test result and through periods of self-isolation: “So, some support around, for the students, around what happens when you get a positive result would be extremely useful to come alongside that [the testing], so that did increase our workload quite considerably.” (P129, male, staff, academic).

#### 3.2.3. Theme 3: Influences on Testing Participation 

(i) Testing Freedom

Considering the challenges faced by participants, several students and staff expressed the motivation to participate in the pilot scheme based on the concept of “testing freedom”. For many, this seemed to be a psychological response of acquiring “peace of mind” from knowing they were not an asymptomatic COVID-19 carrier and they were making efforts to protect their friends, families and local communities. This response then engendered a practical freedom whereby individuals felt that participation in the testing would allow them to visit family members, or made them feel more comfortable to take part in general social activities: “I just wanted to know if I was positive so that I could take steps not to spread it and to kind of know that I was you know at risk and people near me were at risk so that was my main driver” (P128, female, staff, academic). However, the freedom of testing could also have more negative impacts on behaviour, as discussed in theme four. 

(ii) External Influences

Participants referred to many external influences that acted as drivers to their participation in the programme. Many students had been encouraged to take part by family or friends, as well as university staff. Students alluded to being “in it together” and spoke of the expectation from other students that they would take part in the testing “yes, [it’s] like an activity so you would feel left out if you didn’t do it” (P93, female, Yr1, student). Some participants were driven to take part since they viewed this as a mechanism by which university students could contribute to the national effort.

(iii) Curiosity

Some students and staff were motivated to take part in the testing because of an underlying curiosity about their personal health status (i.e., COVID-19 negative or positive, potential immunity to COVID-19), or a scientific interest in the testing processes and the aligned research study.

#### 3.2.4. Theme 4: Testing—Physical and Logistical Factors

(i) Communication

Communication was a pivotal factor throughout the process. Most of the participants referred to the appropriateness and acceptability of communications from the university, including the clear sign-up instructions provided at the outset, concise information about testing and the processes, and appropriate approaches to communication of test results: “yes it was good, we didn’t get told when we had negative results but then we got an email at the end of the day saying that anyone who was positive had been contacted so obviously you knew that you were negative” (P121, female, Yr1, student). Others highlighted problems that had occurred with communications, such as delays in the provision of instructions, and some “less clear” communications that had led to misunderstanding about the purpose of the testing, how to take the swab tests and how samples would be used “I think it would have been better if you got an email every time, just because then you’re kind of like well did they do my test?... There is always a little thing in the back of your head like did it actually test negative? Or did my test not get there?” (P101, female, Yr5, student). Both students and staff had expected to receive their antibody test results but had not received it during the study period, and the lack of communication related to when, or if, results would be available affected adherence to the antibody testing element of the programme. Staff highlighted some inadequacies in university-wide communications about the self-isolation processes and support that students could expect to receive, and the time they spent clarifying communications with students had significantly increased staff workloads. 

(ii) Physical Testing

The act of testing itself was acceptable to the vast majority of participants, and there was no consensus on the preferred methods of testing between saliva or swabs (for presence of SARS-CoV-2) or finger-prick antibody test (for prior exposure). Several students spoke of the efforts staff had made to assist in the process, and video materials, leaflets and explanatory emails were particularly valued to assist with self-testing. A minority referred to negative aspects of the testing (such as swab tests being physically unpleasant to undertake or a fear of needles), but these factors did not appear to deter any of the students from participating and were not reported as reasons for missed tests.

(iii) Practicalities of Testing

No students or staff raised any significant concerns related to the testing processes. There was a general consensus that the frequency of testing was appropriate, and the collection and drop-off locations were convenient, particularly for those who were based on campus: “I thought it was really easy, erm and erm we noticed because the drop off and the collection station is opposite our office, we noticed that the participation from the students… who are living on campus was really, really good.” (P123, female, staff, administrator). Some participants raised practical barriers, such as the logistics of collecting tests when self-isolating, periods of stock depletion which meant swabbing was unavailable, or further challenges related to timetabling, “I think it was pretty much all positive, the only thing was the timings. It was a bit annoying rushing through like some lecture or like running over to put them in before the deadline” (P94, female, Yr1, student). Furthermore, the logistical challenges associated with maintaining adequate supplies of test kits and ensuring all students had access to them, were highlighted by staff.

#### 3.2.5. Theme 5: Testing—Effects on Wellbeing

(i) Improved Mental Wellbeing

The P-ATS was perceived to be extremely important by participants. Despite the negative mental wellbeing impacts of COVID-19 and self-isolation, both students and staff identified positive impacts on wellbeing that were directly associated with the provision of virus testing at the university.

These positive impacts included reassurance about their personal health status, increased perception of safety on campus, reduced anxiety, increased confidence and greater feelings of satisfaction with, and support from, the university during the pandemic: “it was a good kind of confidence boost, you knew that it wasn’t going to spread around university as much as if people were asymptomatic and weren’t being tested. You knew that there was a bit more of a like safety net in a way” (P121, female, Yr1, student).

(ii) Behavioural Change

The positive impacts noted by participants were closely associated with behavioural change. The clear communications had made it easy for students to follow testing instructions and adhere to social distancing guidance on campus. Generally, students and staff felt comfortable with being present on campus while the testing programme was in place. Participants reported feeling happier to visit loved ones and to socialise (in a socially distanced way); final-year students were able to attend rotations (clinical placements), which in some cases were activities that would not have occurred in the absence of testing: “yes I think there has been a few times when I have gone home just for the day to see my family and what not, which I probably wouldn’t have done if the testing wasn’t in place so in that regard it has like changed my behaviour in sort of that way” (P103, female, Yr5, student). Participants who had been required to self-isolate reported that they had been adherent to self-isolation guidance, as had their peers, and they believed that self-isolating was important. Although students did feel more comfortable socialising as a result of the testing programme, those interviewed spoke of the importance of adhering to guidelines and protective behaviours. However, a minority of interviewees had observed a small number of students being less compliant with government COVID-19 restrictions because they were being tested—a negative consequence of “testing freedom”. This frustrated the majority of students who claimed to be compliant: “five out of six of us would be following the same rules anyway to be honest, I would say there is definitely a couple of people that I know that yes are a bit more oh well I am negative so I am just going to do what I want sort of thing” (P102, male, Yr5, student).

## 4. Discussion

### 4.1. Programme Evaluation

Almost all students and staff in this study would take part in an asymptomatic SARS-Cov-2/COVID-19 testing programme again and would recommend it to others. PCR self-testing using throat swab or saliva was highly acceptable (as shown in other community samples) [15]. Testing adherence was high and 4 out of 5 students were confident in their test result. Antibody testing using finger-prick samples was acceptable although lack of communication of antibody test results reduced adherence to finger-prick tests towards programme end. There were no significant problems related to the logistics around the collection of test kits and venues for sample return for students or staff, although there had been a brief period of depleted test kit stock, and one third of first years experienced the occasional difficulty returning the test kit by the required time which was primarily associated with academic timetabling. The process of repeat self-testing was seen to be acceptable to students and staff, and we have demonstrated the acceptability of a mass testing approach over a significantly longer period of time than shown elsewhere [6]. 

Students and staff were largely satisfied with the information received about the testing programme, how information was communicated to them around testing and test kit collection, and the communication of test results. However, in some cases there had been inconsistency in communications from staff to students (e.g., with variations in guidance given to students around self-isolation between those who were operationalising the testing service and academic tutors). One third of participants were dissatisfied with the approach to communicating negative PCR test results taken during the pilot programme although this “batch” approach to communications has been modified since the study end. In our survey, 1 in 5 students reported that they were not confident in their test results, and confidence in test results was related to the number of (antibody) tests completed. This is explained by the qualitative data, which suggests that a lack of confidence in the test results relates to the way in which results had been communicated to individuals by the university during the pilot programme. For example, late or non-receipt of antibody test results, coupled with “batch” communication of negative test outcomes led to uncertainty among students and staff as to whether they had correctly completed the self-testing, or whether their samples had been lost in the laboratory. Timely communication of test outcomes and individual-level communication of all test results may therefore increase confidence in test results, and this may have implications for future testing adherence and COVID-19 vaccine roll-out. 

Moving forwards, it may be useful to review and standardise the communication plan for the provision of guidance around the testing processes, test results and self-isolation, taking into account the importance of student and staff mental wellbeing and perceived risk, as well as the impact of specific wording used in communications which is known to influence individuals’ understanding of health test results [16]. More broadly, communications have been shown to be critical, since perceived sufficiency of information provided has been shown to influence anxiety and behavioural responses to COVID-19 [17], as well as other pandemics (e.g., influenza) [18]. However, the volume of information per communication may be important, since our qualitative findings highlight students’ desires for information to be reiterated due to receipt of over-length emails which, for some, resulted in them missing information. Providing a clear timeframe for communication of all test results (in our case, the antibody tests) would reassure students and reduce their anxiety, which may consequently impact their future testing behaviour. 

Many students reported that they gained new knowledge about COVID-19 and testing procedures from taking part, particularly students in their first year. Although students who had self-isolated at some point during the pilot appeared to be more adherent to the testing (i.e., completed on average more tests), this should be interpreted with caution, due to limited statistical power and restricted sampling from the target population.

### 4.2. Adherence to Testing

Students with lower levels of anxiety and greater satisfaction with university communications around P-ATS were more likely to adhere to PCR and antibody tests. Students who were dissatisfied with university communications were less adherent to PCR testing, as were those with higher anxiety. Anxiety levels in our higher education sample were higher than those detected in adult samples globally (21%) [19]. This further highlights the importance of efforts to protect mental wellbeing in university students during the implementation of national and local containment measures. This is supported by data from the staff focus group, where protecting the mental wellbeing of students in self-isolation was deemed to be essential to the success of future testing programmes. 

The findings clearly indicate the need for clarity and the consistency of communications around testing approaches, test outcomes and self-isolation, promotion of positive wellbeing, and support for mental health during self-isolation and more broadly. Experiences of confinement during a pandemic can negatively impact the psychological wellbeing of young adults, including those in the general population [20] and specifically, those in a higher education context [21,22,23]. However, a one-size-fits-all approach to mental health support in this context is likely to be insufficient due to disparities in mental health outcomes of higher education students during the pandemic, with those in the health professions, younger and more affluent students faring better than other student groups [20]. Increased adherence to testing was also associated with higher perceived risk of COVID-19 to themselves (PCR test) or friends and family (antibody test). This supports a previous study which demonstrated a relationship between perceived COVID-19 risk and adverse mental health outcomes in a UK community sample earlier in the COVID-19 pandemic [24].

### 4.3. Risk Perceptions and Protective Behaviours

Students and staff generally perceived that their safety on campus was increased due to university asymptomatic testing. Increased perception of safety did not appear to reduce adherence to social distancing or self-isolation, but some interviewees reported observing a minority of students failing to comply to university advice. There was a general perception that protective behaviours (social distancing, handwashing, face masks and self-isolation) were very important, but students’ views on the importance of protective behaviours were not associated with their perceived risk of COVID-19, which seemed to relate more to their personal experiences of COVID-19 (e.g., self, friends or family) and whether they had been required to self-isolate previously. Similarly, staff alluded to the influence of personal experiences of COVID-19 on their decisions to take part in the P-ATS. Students who chose not to participate in the P-ATS perceived their risk of COVID-19 to be lower compared with students who took part, although low statistical power means this should be interpreted with caution.

### 4.4. Study Strengths and Limitations

To our knowledge, this is the first study to explore in depth, the perceptions and experiences of students and staff following delivery of a mass SARS-Cov-2/COVID-19 testing programme in a university setting. This study demonstrates the perceived value of the mass testing approach to students and staff in a higher education setting, although the economic and health impact of this approach is yet to be established. Study findings will inform future deployments of COVID-19 virus and antibody testing on university campuses and may provide insights to inform the roll-out of COVID-19 vaccines in the future. However, these findings should be considered in the context of the environment in which this pilot deployment was delivered (a single campus of a multi-campus university, in a semirural location), at a time of frequent changes in patterns of virus transmission. The data were collected prior to the second surge of COVID-19 in the UK at which time the participating university deployed mass testing more broadly to include students living on its other campuses closer to the city. In these subsequent deployments, a large number of positive cases were detected, leading to high numbers of students self-isolating, and this was coupled with lower uptake of mass testing among students. Therefore, our findings may not be directly transferable to different settings or across rapidly changing national and local contexts. The reasons for the subsequent reduction in testing uptake, apparent changes in students’ attitudes to testing and the barriers and enablers of self-isolation need to be explored. This is particularly pertinent in the light of the government’s adoption of a phased return of students to UK universities in January 2021, whereby students returning to campus first may be subject to successive waves of virus exposure bringing the possibility of needing to self-isolate more than once. 

Validity was strengthened as data were collected and analysed by researchers who were not involved in the delivery of the testing programme. There were more female than male participants in our study (no students identified as non-binary), which reflects the gender balance of students completing a veterinary degree, with proposed figures of 77% [25] and 80% [26], but is higher than the proportion of females across all higher education students in the UK, estimated to be 57% [27]. Due to the cross-sectional survey data collected in this study, it is not possible to determine the temporal nature of any associations presented here (i.e., whether the P-ATS led to any psychological or behavioural changes). The small sample size may affect the generalisability of results, although the survey response rate was adequate to address the study aims. Similarly, given the aim of the study, the sample specificity, the rich dataset, in-depth insights into the phenomena of interest and the analysis approach adopted [28], the qualitative sample was deemed to have sufficient information power, although further insights from a larger sample of staff may be valuable for future research. The positive evaluation should be interpreted in the light of known drawbacks of universal testing such as false-positive and false-negative tests, the difficulty of defining an active infection and significant resource implications [29,30].

## 5. Conclusions

University students and staff want to keep campuses safe and contribute to the national effort to prevent and manage outbreaks of COVID-19. Asymptomatic COVID-19 testing is highly acceptable to students and staff in a university campus setting, using two types of PCR test (swab or saliva) and finger-prick antibody tests. Adherence to testing is higher for students with prior experience of self-isolating. Testing adherence is directly related to lower anxiety and students’ satisfaction with communications from the university around testing and associated support. Student anxiety levels are largely associated with personal worries about themselves or their families contracting the virus, and their satisfaction with university COVID-19 communications. Adequacy of support for student mental wellbeing will be critical during and after the pandemic, but the implications for workload and emotional burden on university staff should be carefully considered. Uptake, adherence and satisfaction with mass asymptomatic testing services in a university setting will be influenced by the continued support and time investment from university staff who have direct contact with students, the clarity of communications particularly around test results, and the level of practical, welfare and emotional support provided to students who are self-isolating.

## Figures and Tables

**Figure 1 ijerph-18-00188-f001:**
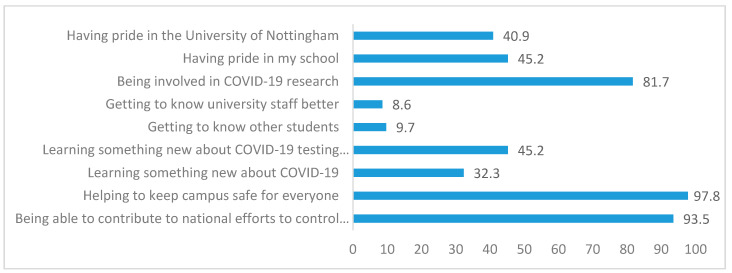
Perceived benefits of university testing service (*n* = 99, % yes).

**Figure 2 ijerph-18-00188-f002:**
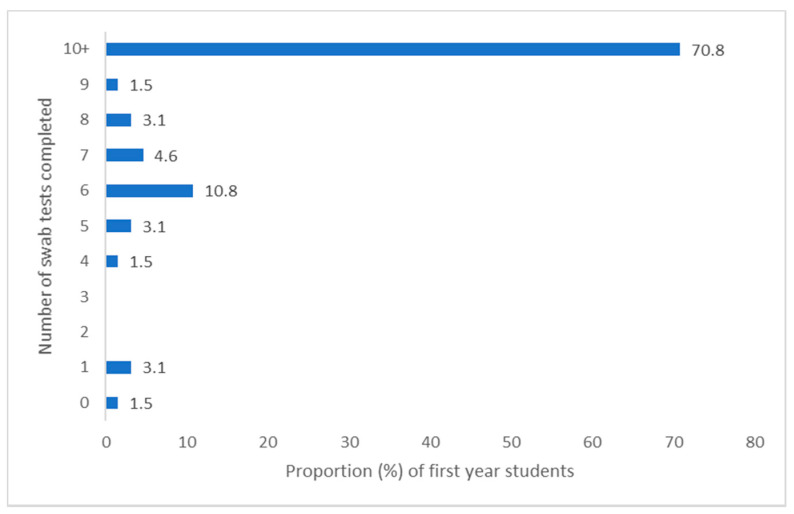
Proportion of year one students completing between 0–10 swab tests during P-ATS.

**Figure 3 ijerph-18-00188-f003:**
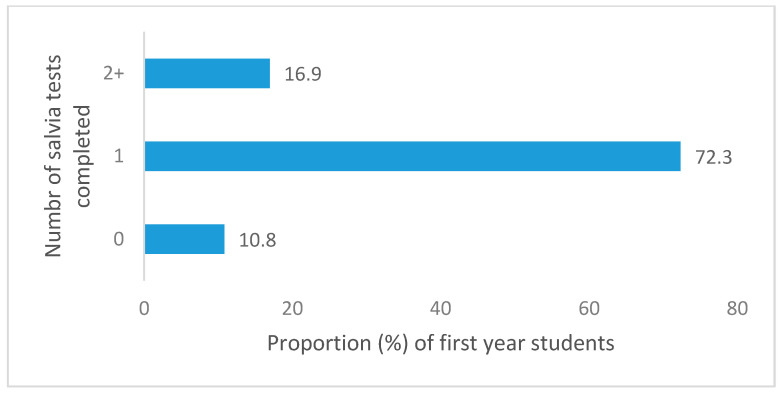
Proportion of year one students completing between 0–2+ saliva tests during P-ATS.

**Figure 4 ijerph-18-00188-f004:**
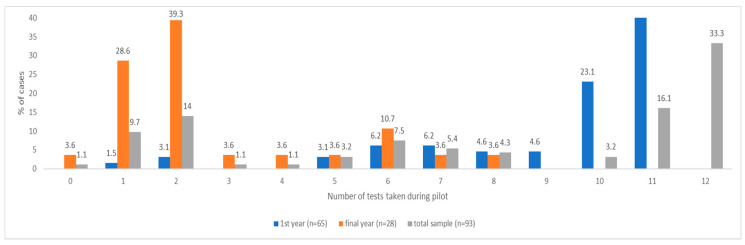
Reported frequency of number of PCR tests (swab and saliva) completed across 12 weeks of testing during P-ATS by total sample and stratified by first and final years.

**Figure 5 ijerph-18-00188-f005:**
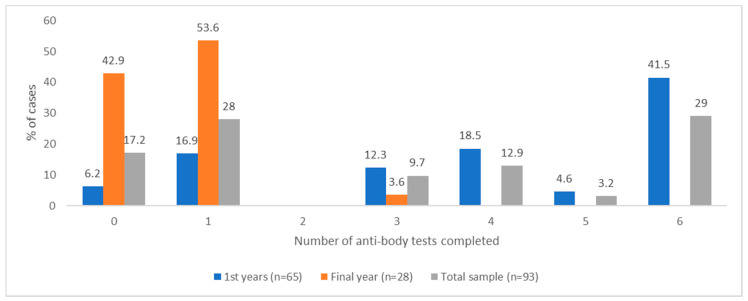
Reported frequency of number of antibody tests completed across 12 weeks of testing during P-ATS by total sample and stratified by first and final years.

**Table 1 ijerph-18-00188-t001:** Sample characteristics for interview and focus group participants.

ID	Gender	Year of Study	Ind/SG/FG †	Date	Time	Duration (mins)
P101	F	5	IND	16 October 2020	16:30	36:31
P106	F	5	IND	18 October 2020	18:00	44:56
P116	F	1	IND	24 October 2020	19:00	48:33
P112	F	1	IND	24 October 2020	10:30	17:50
P117	F	1	IND	25 October 2020	14:30	50:47
P121	F	1	IND	30 October 2020	11:00	13:12
P91	F	1	SG	02 September 2020	14:00	32:50
P92	F	1	SG	02 September 2020	14:00	32:50
P93	F	1	SG	02 September 2020	14:00	32:50
P99	M	1	SG	14 October 2020	14:00	32:50
P100	M	1	SG	14 October 2020	14:00	32:50
P104	F	1	SG	17 October 2020	16:30	45:30
P105	M	1	SG	17 October 2020	16:30	45:30
P102	M	5	SG	17 October 2020	10:30	43:55
P103	F	5	SG	17 October 2020	10:30	43:55
P107	F	1	SG	18 October 2020	11:30	42:06
P108	M	5	SG	18 October 2020	11:40	42:06
P109	F	5	SG	23 October 2020	16:30	48:19
P110	F	1	SG	23 October 2020	16:30	48:19
P111	F	1	SG	23 October 2020	16:30	48:19
P113	F	5	SG	24 October 2020	16:30	48:48
P114	F	5	SG	24 October 2020	16:30	48:48
P115	F	5	SG	24 October 2020	16:30	48:48
P118	F	1	SG	27 October 2020	19:00	48:51
P119	F	5	SG	27 October 2020	19:00	48:51
P120	F	1	SG	27 October 2020	19:00	48:51
P94	F	1	FG	25 September 2020	13:00	53:49
P95	F	1	FG	25 September 2020	13:00	53:49
P96	F	1	FG	25 September 2020	13:00	53:49
P97	F	1	FG	25 September 2020	13:00	53:49
P98	F	1	FG	25 September 2020	13:00	53:49
Staff Participants						
ID	Gender	Job Role	Ind/SG/FG †	Date	Time	Duration (mins)
P127	F	Academic	FG	28 October 2020	15:00	70:58
P129	M	Senior Academic	FG	28 October 2020	15:00	70:58
P130	F	Academic	FG	28 October 2020	15:00	70:58
P126	F	Clinical academic	FG	28 October 2020	15:00	70:58
P123	F	Senior administrator	FG	28 October 2020	15:00	70:58
P128	F	Academic	FG	28 October 2020	15:00	70:58
P125	F	Academic	FG	28 October 2020	15:00	70:58
P124	F	Clinical academic	FG	28 October 2020	15:00	70:58
P122	F	Administrator	FG	28 October 2020	15:00	70:58
P131	M	Senior Academic	FG	28 October 2020	15:00	70:58

† Ind: Individual Interview; SG: Small Group Interview (2–3 people); FG: Focus Group Interview (4+ people).

**Table 2 ijerph-18-00188-t002:** Sample characteristics of survey respondents (*n* = 99).

Respondent Characteristics (*N* = 99)	Category	*n* (%)
Year of study	First year	69 (69.7)
	Final year	30 (30.3)
Gender		
	Male	13 (13.1)
	Female	84 (84.8)
	Prefer not to say	2 (2.0)
Ethnicity		
	Asian or Asian British	3 (3.03)
	White	91 (93.9)
	Prefer not to say	3 (3.0)
Student background		
	Home student	91 (91.9)
	European student (EU)	8 (8.1)
Accommodation during term	Private accommodation with family	4 (4.0)
	Private accommodation with others	32 (32.3)
	Halls of residence	63 (63.6)

**Table 3 ijerph-18-00188-t003:** Anxiety in study sample (*n* = 99).

		Mean (SD)	Group Mean Difference
Total sample	*n* = 99	7.21 (6.01); Range: 0–21	
Year of study	First year (*n* = 69)	6.73 (6.02)	n.s
	Final year (*n* = 30)	8.33 (5.92)	
Students’ self-isolating	Yes (>=once; *n* = 36)	7.39 (6.41)	n.s
	No (*n* = 63)	7.11 (5.82)	
Positive cases	Yes (*n* = 4)	6.50 (5.00)	
	No (*n* = 89)	7.18 (6.00)	n.s.
Accommodation during term	Private accommodation (*n* = 36)	8.89 (6.27)	* *t* (97) = 2.138, *p* = 0.035, Cohen’s d = 0.44
	Halls of residence (*n* = 63)	6.25 (5.68)	
P-ATS pilot	Pilot participants (*n* = 93)	7.15 (6.00)	n.s.*
	Non-pilot participants (*n* = 6)	8.17 (7.52)	

Note: Group mean differences in GAD-7 were examined using an independent *t* test. * Due to limited statistical power test results should be interpreted with caution (power = 0.55).

**Table 4 ijerph-18-00188-t004:** Experiences of the P-ATS within and across study sample (total, *n* = 99; first years, *n* = 69; final years, *n* = 30).

Pilot Procedure or Participant Experience	Response Options	Antibody Tests *n* (%)			Swab and Saliva Tests † *n* (%)		
		Total Sample	First Year	Final Year	Total Sample	First Year	Final Year
How did you collect test kits?	Collected myself	32 (34.4)	18 (27.7)	14 (50.0)	36 (38.7)	17 (26.2)	19 (67.9)
	Collected by others in household/bubble	12 (12.9)	11 (16.9)	1 (3.6)	9 (9.7)	8 (12.3)	1 (3.6)
	Mix of both	36 (38.7)	35 (53.8)	1 (3.6)	46 (49.5)	39 (60.0)	7 (25.0)
	Didn’t collect any test	13 (14)	1 (1.5)	12 (42.9)	2 (2.2)	1 (1.5)	1 (3.6)
Were you able to return the completed test kit by requested time?	Yes, always	58 (62.4)	42 (64.6)	16 (57.1)	61 (65.6)	39 (60.0)	22 (78.6)
	Yes sometimes	19 (20.4)	19 (29.2)	-	31 (33.3)	26 (40.0)	5 (17.9)
	Didn’t complete any test	16 (17.2)	4 (6.2)	12 (42.9)	1 (1.1)	-	1 (3.6)
Were you satisfied with location of the drop-off?	Yes	74 (79.6)	59 (90.8)	15 (53.6)	88 (94.6)	63 (96.9)	25 (89.3)
	No	2 (2.2)	1 (1.5)	1 (3.6)	-	-	1 (10.7)
	Didn’t drop off a completed kit myself	17 (18.3)	5 (7.7)	12 (42.9)	5 (5.4)	2 (3.1)	-
† Satisfied with the university approach to communicating positive test results	Yes	-	-	-	86 (92.5)	59 (90.8)	27 (96.4)
† Satisfied with the university approach to communicating negative test results	Yes	-	-	-	61 (65.6)	49 (75.4)	12 (42.9)

† Positive and negative PCR (swab and saliva) test results were communicated to students during the study period; antibody test results had not been distributed by week 12.

**Table 5 ijerph-18-00188-t005:** Overview of satisfaction with P-ATS and uptake of future testing services (%).

Item	Response Option	Total Sample (*n* = 99)	First Year (*n* = 69)	Final Year (*n* = 30)
Would take part in University testing in future if it was offered to me	Yes	94 (94.9)	66 (95.7)	28 (93.3)
Would encourage others to take part in University testing	Yes	97 (98.0)	97 (97.1)	30 (100)
		P-ATS sample (*n* = 93)	P-ATS First year (*n* = 65)	P-ATS Final year (*n* = 28)
Satisfaction with information received	Very satisfied	43 (46.2)	34 (52.3)	9 (32.1)
	Somewhat satisfied	46 (49.5)	29 (44.6)	17 (60.7)
	Somewhat dissatisfied	4 (4.3)	2 (3.1)	2 (7.1)
	Very dissatisfied	-	-	-
Satisfaction with programme communications	Very satisfied	43 (46.2)	34 (52.3)	9 (32.1)
	Somewhat satisfied	40 (43)	26 (40.0)	14 (50.0)
	Somewhat dissatisfied	9 (9.7)	4 (6.2)	5 (17.9)
	Very dissatisfied	1 (1.1)	1 (1.5)	-
Gained new knowledge	Yes	45 (51.6)	39 (60.0)	9 (32.1)

**Table 6 ijerph-18-00188-t006:** Correlation matrix of frequency of PCR test completion (combined swab/saliva), mental health, and programme satisfaction (*n* = 91).

	1	2	3	4	5	6	7	8	9
1. Frequency of COVID-19 test completion	1	−0.179	−0.287 **	*−0.239* *	*−0.001*	*−0.267* *	−0.282 **	−0.344 **	−0.189
2. Worry about getting COVID-19		1	0.565 **	*−0.026*	*−0.173*	*0.04*	−0.023	0.141	−0.066
3. Worry about friends and family getting COVID-19			1	*−0.009*	*−0.19*	*0.092*	0.005	0.124	0.163
4. Satisfaction with drop-off location +				1	*0.292* **	*0.125*	*0.253* *	*0.177*	*0.153*
5. Satisfaction with positive test result communication +					1	*0.133*	*0.281* **	*0.311* **	*0.051*
6. Satisfaction with negative test result communication +						*1*	*0.293* **	*0.354* ***	*0.024*
7. Satisfaction with overall P-ATS information received							1	*0.564* **	*0.058*
8. Satisfaction with how overall P-ATS information was communicated								1	0.016
9. Gained new knowledge through the pilot									1

*** *p* < 0.001, ** *p* < 0.01, * *p* < 0.05; *n* = 91; italic correlation coefficients indicated non-parametric correlations. Two cases removed due to standardised residual beyond +/− 3.0. + Binary variables coded 0 for “yes” and 1 “no”.

**Table 7 ijerph-18-00188-t007:** Correlation matrix of frequency of antibody test completion, mental health, and programme satisfaction (*n* = 93).

	1	2	3	4	5	6	7	8
1. Frequency of antibody tests completed	1	−0.071	−0.220 *	−0.226 *	−0.318 **	−0.169	*−0.137*	*−0.533* **
2. Worry about getting COVID-19		1	0.533 **	−0.014	0.148	−0.065	*−0.042*	*0.152*
3. Worry about friends and family getting COVID-19			1	0.023	0.139	0.159	*−0.131*	*0.216* *
4. Satisfaction with overall P-ATS information received				1	0.608 **	0.07	*0.317* **	*0.259* *
5. Satisfaction with how overall P-ATS information was communicated					1	0.015	*0.272* **	*0.403* ***
6. Gained new knowledge through P-ATS						1	*−0.01*	*0.194*
7. Confidence in test outcomes +							1	0.013
8. Satisfaction with drop-off location +								1

*** *p* < 0.001, ** *p* < 0.01, * *p* < 0.05; *n* = 91; italic correlation coefficients indicated nonparametric correlations. + Binary variables, yes coded “0” and no “1”.

**Table 8 ijerph-18-00188-t008:** Summary of hierarchical regression analysis of variables predicting number of PCR (swab/saliva) tests taken.

		Frequency of Swab Tests Completed		
Step	Predictor	β	B	Basis
Step 1				
(force wise)	Gender	−0.088	−0.815	0.538
	Year of Study	−0.848 ***	−1.952	0.134
	∆R^2^			0.707 ***
Step 2				
(stepwise)	Gender	−0.070	−0.648	0.513
	Year of Study	−0.821 ***	−1.891	0.129
	Anxiety (GAD-7)	−0.180 **	−0.129	0.040
	∆R^2^			0.031 **
Step 3				
(stepwise)	Gender	−0.115 *	−1.062	0.499
	Year of Study	−0.868 ***	−1.999	0.125
	Anxiety (GAD-7)	−0.296 ***	−0.213	0.045
	Worry about getting COVID-19	0.227 **	1.384	0.400
	∆R^2^			0.032 **
Step 4				
(stepwise)	Gender	−0.098	−0.907	0.494
	Year of Study	−0.846 ***	−1.949	0.125
	Anxiety (GAD-7)	−0.244 ***	−0.175	0.047
	Worry about getting COVID-19	0.205 **	1.254	0.397
	Satisfaction with the way information was communication	−0.118 *	−0.723	0.344
	∆R^2^			0.011 *
	(Constant)		17.623 ***	1.22

*** *p* < 0.001, ** *p* < 0.01, * *p* < 0.05, *n* = 91; two cases removed as standard residuals exceed +/− 3.5; (model 5) R^2^ = 0.821, adjusted R^2^ = 0.808. Excluded variables: worry about friends and family getting COVID-19, satisfaction with way a positive test was communicated, satisfaction with way a negative test was communicated satisfaction with drop-off points, satisfaction with information received during programme, confidence in outcome of test, and gained new knowledge through taking part in pilot.

**Table 9 ijerph-18-00188-t009:** Summary of hierarchical regression analysis of variables predicting number of antibody tests taken (*n* = 93).

		Frequency of Antibody Tests Completed		
Step	Predictor	β	B	Basis
Step 1				
(forced entry)	Gender +	0.052	0.263	0.405
	Year of Study	−0.648 ***	−0.832	0.103
	∆R^2^	0.431 ***		
Step 2				
(stepwise)	Gender	0.082	0.415	0.388
	Year of Study	−0.609 ***	−0.781	0.099
	Anxiety (GAD-7)	−0.247 **	−0.099	0.031
	∆R^2^	0.059 **		
Step 3				
(stepwise)	Gender	0.021	0.109	0.374
	Year of Study	−0.674 ***	−0.865	0.096
	Anxiety (GAD-7)	−0.413 ***	−0.165	0.034
	Worry about getting COVID-19	0.326 ***	1.111	0.306
	∆R^2^	0.066 ***		
Step 4				
(stepwise)	Gender	0.021	0.108	0.363
	Year of Study	−0.590 ***	−0.757	0.102
	Anxiety (GAD-7)	−0.345 ***	−0.137	0.035
	Worry about getting COVID-19	0.296 ***	1.010	0.300
	Satisfaction with the location of drop-off location	−0.203 *	−0.615	0.243
	∆R^2^	0.03 *		
Step 5				
(stepwise)	Gender	0.078	0.395	0.382
	Year of Study	−0.584 ***	−0.749	0.100
	Anxiety (GAD-7)	−0.310 ***	−0.124	0.035
	Worry about getting COVID-19	0.257 **	0.876	0.301
	Satisfaction with the location of drop-off location	−0.205 8	−0.620	0.239
	Confidence in outcome test result ++	−0.153 8	−0.895	0.431
	∆R^2^	0.02 *		
	(Constant)		6.22	0.938

*** *p* < 0.001, ** *p* < 0.01, * *p* < 0.05, *n* = 93; (model 5) R^2^ = 0.606, adjusted R^2^ = 0.579; Excluded variables: worry about friends and family getting COVID-19, satisfaction with information received during programme, satisfaction with the way information was communication, and gained new knowledge through taking part in pilot. + male is the referent group. ++ “yes” is the referent group.

**Table 10 ijerph-18-00188-t010:** Summary of hierarchical regression analysis of variables predicting anxiety (GAD-7).

		Anxiety (GAD-7)		
Step	Predictor	β	B	Basis
Step 1				
(forced entry)	Gender +	0.122	1.544	1.323
	Year of study	0.16	0.515	0.335
	∆R^2^	0.036		
Step 2				
(stepwise)	Gender	−0.017	−0.219	2.776
	Year of Study	0.002	0.005	1.163
	Worry about getting COVID-19	0.560 ***	4.799	0.298
	∆R^2^	0.275 ***		
Step 3				
(stepwise)	Gender	−0.007	−0.091	1.124
	Year of Study	−0.07	−0.226	0.3
	Worry about getting COVID-19	0.410 ***	3.508	0.911
	Worry about friends and family getting COVID-19	0.294 **	2.37	0.87
	∆R^2^	0.054 **		
Step 4				
(stepwise)	Gender	−0.654	1.05	−0.051
	Year of Study	−0.412	0.281	−0.128
	Worry about getting COVID-19	3.361	0.843	0.393
	Worry about friends and family getting COVID-19	2.28	0.805	0.283
	Satisfied with the way in which information was communicated to me.	2.743	0.691	0.323
	∆R^2^	0.098 ***		
	(Constant)		2.047 **	3.086

R^2^ = 0.462, adjusted R^2^ = 0.431; Note: *n* = 93, *p* < 0.05, ** *p* < 0.01, *** *p* < 0.001; Excluded variables: overall satisfaction with information received through programme, gained knowledge through the programme, confidence in the test outcome. + male is the referent group.

**Table 11 ijerph-18-00188-t011:** Mean (standard deviation) of perceived risks and importance of health protective behaviours within and across study groups.

	Total Sample (*n* = 99)	Pilot Participants (*n* = 93)	Nonparticipants (*n* = 6)	Group Difference ^†^	Negative Test (*n* = 89)	Positive Test (*n* = 4)	Group Difference ^†^	Have self-Isolated (*n* = 36)	Have not Self-Isolated (*n* = 63)	Group Difference ^†^
Perceived importance of protective behaviours for virus control (rated 1–10)										
Social distancing	8.24 (1.79)	8.29 (1.75)	7.50 (2.58)	n.s.	8.33 (1.74)	7.5 (2.08)	n.s.	7.89 (1.98)	8.44 (1.64)	n.s.
Regular hand washing	9.04 (1.47)	9.02 (1.50)	9.33 (1.033)	n.s.	9.03 (1.52)	8.75 (.96)	n.s.	8.69 (1.64)	9.24 (1.34)	n.s.
Self-isolating	9.04 (1.47ß)	9.10 (1.714)	8.83 (1.329)	n.s.	9.16 (1.57)	7.75 (3.86)	n.s.	8.61 (*n* = 2.19)	9.35 (*n* = 1.27)	n.s.
Wearing a face covering	9.08 (1.69)	8.39 (2.08)	7.17 (1.722)	n.s.	8.40 (2.06)	8.00 (2.83)	n.s.	8.25 (2.10)	8.35 (2.7)	n.s.
Perceived risk of COVID-19 (pre-pilot; rated 1–10)	5.56 (1.768)	5.67 (1.71)	3.83 (1.94)	*t* (92) = 2.53, *p* < 0.05	5.69 (1.72)	5.25 (1.50)	n.s.	5.42 (1.99)	5.63 (1.64)	n.s.
Perceived risk of COVID-19 (post-pilot; rated 1–10)	6.47 (2.01)	6.54 (2.00)	5.50 (2.08)	n.s.	6.45 (1.98)	8.50 (1.29)	*t* (91) = 2.043, *p* < 0.05	6.22 (2.28)	6.62 (1.85)	n.s.

^†^ Independent *t* test conducted. However, these statistics are underpowered (<0.8) and should be interpreted with caution, due to elevated risk of a Type II error.

**Table 12 ijerph-18-00188-t012:** Key themes and subthemes and their representative quotes.

Theme	Subthemes	Representative Quotations
Emotional Responses to COVID-19	Negative Feelings	I am definitely suffering from COVID fatigue, that is how much I think about COVID (P126, Female, Staff—Academic) I think it has been a lot of like adaptation but I think the most stressful part was when I came back and had to go on rotations because then you couldn’t avoid the thing that was stressing you, which was like seeing other people. (P101, Female, Yr5, Student).
	COVID-19 Around Me	My eldest son has Asthma and I mean he is nine, so he is young, but it was still a concern that I was thinking crikey if he gets COVID is that going to be a massive deal? (P128, Female, Staff—Academic) My parents are both over 60, my mother in particular was shielding, I opted to stay at university, both for exams and for lockdown purposes (P106, Female, Yr5, Student) I had not left my house since March and so I didn’t know anyone who had had COVID or been out anywhere that I could get COVID really, so I was quite nervous moving here (P111, Female, Yr1, Student)
	Coping with COVID-19	Sometimes I feel so much better if I just don’t watch the news and not even think about it because you turn the news on for a bit in the morning and it is just a bit depression” (P91, Female, Yr1, Student) Yes, I think isolation was all right, it was just erm trying to keep busy really. Not get too bored (P105, Male, Yr1, Student)
University Life during COVID-19	Adapting to a new normal	You kind of get used to studying online and then you go there, and you get used to studying there, and then you come back and then you’re thinking like you are virtually (P117, Female, Yr1, Student) If you’re in a household where you don’t really get along, or you don’t really socialise and you have the option to go and see I don’t know the football team or some friends that you went to university with, or somebody is having a get together and you don’t want to feel left out. It is much more challenging to say no in those types of situations (P106, Female, Yr5, Student) Because a lot of my release is going outside, riding horses and just going for walks and stuff and when you can’t do that, and you don’t really even have a window to lean out of. It drives you a bit nuts (P107, Female, Yr1, Student)
	Improving University Life	Means of having food delivered is a big one, potentially access to some sort of entertainment, be it you know a subscription package or… or like just some form of entertainment. Additionally I think they are the big ones, people get bored and people feel that they need to go out and get stuff and if you have that then… the people that will adhere would adhere much more happily (P106, Female, Yr5, Student) Just helping to make the university a safer place and just keeping, erm, COVID levels as low as possible by making people isolate… I think all we need is like shopping deliveries and any post etcetera. I think that is all in place as it is (P112, Female, Yr1, Student)
Influences on Testing Participation	Testing Freedom	I just wanted to know if I was positive so that I could take steps not to spread it and to kind of know that I was you know at risk and people near me were at risk so that was my main driver (P128, Female, Staff—Academic) I have opted just to do the testing for my own peace of mind (P106, Female, Yr5, Student)
	External Influences	I guess you could say I was influenced by the university really pressing us to do it as a good idea (P100, Male, Yr1, Student) I did it because all of my flat did it and we just decided that we would do it together, erm and also I just wanted to help out and be part of the research (P112, Female, Yr1, Student)
	Curiosity	We are scientists really and I think we should be doing these things, we should be pushing ourselves, we should be seeing what we can do and how we can do it, so curiosity, but also you know we were the very first people who actually designed something like that and went through the pain because there was loads of pain on behalf of people who are doing it. It is not you know it wasn’t that easy to take off, so I think that is something to be proud of. (P126, Female, Staff—Academic) I thought the study [research] was quite interesting so I guess the interest would drive me to continue doing the study. (P94, Female, Yr1, Student)
Testing—Physical and Logistical Factors	Communication	Yes, it was good, we didn’t get told when we had negative results but then we got an email at the end of the day saying that anyone who was positive had been contacted so obviously you knew that you were negative. Erm which I think worked absolutely fine like I don’t think you need to be notified if you’re negative if there is a lot of people doing the study, I guess it takes a lot of time. (P121, Female, Yr1, Student) I think it would have been better if you got an email every time, just because then you’re kind of like well did they do my test? Or did I just… like you’re just unsure there is always a little thing in the back of your head like did it actually test negative? Or did my test not get there? (P101, Female, Yr5, Student) I don’t know about antibody’s but with regards to that I… we still haven’t heard. Antibody testing, I did the one in the second week, I didn’t do the first one and I didn’t do the last antibody but I did every other one and I haven’t got a clue if I had or haven’t had COVID at any point. I think they are lost in the system somewhere (P116, Female, Yr1, Student)
	Physical testing	I thought it all worked very well and I know some people were worried about sticking swabs in various different places, I didn’t think there was any problem with that at all (P129, Male, Staff—Academic) I personally felt that actually, erm, doing different tests was very useful for students, because they will understand how the animals feel when they are having certain things done so that was definitely something which I even considered that they should be doing (P126, Female, Staff—Academic) You can’t really get around the whole finger pricking thing because I know some people just don’t like the whole needle, getting stabbed aspect of that but the spitting in to a tube is kind of disgusting but I think it is easier than like the whole swab in the back of your throat (P91, Female, Yr1, Student)
	Practicalities of testing	The location was convenient because it was sort of on our way to most of our practical sessions so if we happened to have a practical that day we could drop them off on the way... I think it was pretty much all positive, the only thing was the timings. It was a bit annoying rushing through like some lecture or like running over to put them in before the deadline” (P94, Female, Yr1, Student) The only issue that we had with it was like if we were on rotations, sometimes the tests wouldn’t be brought to the rotation site because they were supposed to be, but by our vet school they were supposed to deliver them to the rotation site because you couldn’t go to uni, do the test, drop it off and still get to your rotation on time. So that was the only kind of issue we had with it (P101, Female, Yr5, Student) I think once a week was OK to be fair (P121, Female, Yr1, Student) I think the weekly thing was just about right (P100, Male, Yr1, Student)
Testing—Effects on Wellbeing	Improved Mental Wellbeing	I think it is just for kind of peace of mind it helped a bit, not having to worry about it all of the time and everyone said oh you’re going to university are you not worried? It was just kind of nice to know that there is awareness, and it is not just you know social distancing, you are actively trying to help as well I think. (P92, Female, Yr1, Student) I think [name 6] I just found it hugely reassuring, I was really, really keen to take part when I heard about that it was going to happen and I just… it made me feel well yes just that word it was just really reassuring to know that I was getting regularly tested and in a way I know you can’t directly say that everybody in my immediate family is OK but it was almost like I could act like the canary going down the mine and that there was a certain amount of reassurance as well that if I was negative there was a high chance that my children and my husband were also negative (P124, Female, Staff—Academic)
	Behavioural change	I mean I guess testing negative maybe made me a bit more comfortable to go to like go and play football and things like that but I am not a massive sort of go out person anyway. I was only really sort of the gym and that so… I guess it is a more sort of like oh it is all right I can go play football with a group of people and stuff but again I think… it was like the prospect of a potential positive test the week after so I think overall I probably stayed about the same really. (P105, Male, Yr1, Student) It makes you kind of think more about who you’ve been in contact with, give people [unclear 15:04] tested positive then you’re suddenly a bit more aware of who you have met up with that week and stuff. Making sure that you are within the guidelines and not seeing more then the number of people you can be meeting with or households and things. Just in case (P104, Female, Yr1, Student) Yes, and if there was any social events on and if there was any lectures and stuff, I would just say well you can’t come in until you have had the test. I don’t know that sounds really and it is difficult because you can’t make someone have it but at the same time, why should that person put everyone else at an increased risk? (P102, Male, Yr5, Student)

## Data Availability

The data presented in this study are available in File S6: PAT-S Survey Data.

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
