# Peer review of "Perceptions and Experiences of the University of Nottingham Pilot SARS-CoV-2 Asymptomatic Testing Service: A Mixed-Methods Study"

_ijerph, 2020, doi:10.3390/ijerph18010188_

Round 1

Reviewer 1 Report

I had the opportunity to review the manuscript, “Perceptions and experiences of the University of Nottingham pilot asymptomatic testing service: A mixed-methods study.” The purpose of the manuscript was to explore student and staff perceptions and experiences of a pilot COVID-19 asymptomatic testing service. Participants were mostly-female college students who completed surveys and focus groups.

This manuscript is timely and addresses pressing global issues. The study focuses on multiple perceptions of the testing process (uptake, adherence, acceptability), which provides important breadth. Overall, I am enthusiastic about this manuscript and believe it can make a valuable contribution to addressing pandemic-related issues.

Nonetheless, there are several limitations to the manuscript in its current form. These are detailed below, along with suggestions to address them where applicable.

  • P .3 — “Participation in P-ATS was on a voluntary basis and aimed to be complementary to the testing strategy in the United Kingdom (UK).” Please provide additional information about the national testing strategy and how these two programs complement each another. People unfamiliar with the UK testing strategy (or any national testing strategy…) will have trouble understanding how these programs fit together.
  • The authors first note that recruitment was primarily targeted to students in their first year who were living in university accommodation on campus, but then they state: “Sampling was purposive, to include students of any gender, those living on or off campus, and those in their first or final year of study.” These statements seem contradictory; please clarify.
  • On a related note, throughout the manuscript, distinction is repeatedly made between first and final year students. Aside from year in school, are there noteworthy differences between these groups that warrant comparison? I am unfamiliar with schooling in the UK and this program in particular, so I might be ignorant to important differences, but I am unsure of the value of these comparisons—especially in light of the relatively underpowered sample.
  • 3 — “Recruitment for the students continued until no new knowledge was being obtained indicating thematic saturation.” How was this determined? Were data coded and analyzed as it was received? If so, that seems unusual from a methodological perspective—and difficult to identify themes across participants in the thematic analysis.
  • I recognize the burden of data coding, but only one person (LB) coded most of the data and generated themes. A second researcher coded only four transcripts, which is quite low. Thus, data analysis and theme generation were primarily conducted by one person. Given how central (and extension) this data analysis is to the paper, it is perplexing that the identified themes primarily represent one person’s judgment. Relatedly, the authors note that “themes were cross-checked and agreed by a third member of the research team (HB).” What does it mean that “themes were cross-checked”? How were discrepancies between coders identified and resolved? More detail is needed.
  • I appreciate the various group comparisons, but most (all?) are under-powered, given the small simple size, as the authors note. Presenting them is somewhat misleading; nonsignificant comparisons might have a meaningful effect if there was sufficient statistical power to detect it, and significant effects may be an artifact of sample nuances (e.g., uneven groups: N= 36 and 63).
  • Table 5 is difficult to interpret; a figure might be better. Figures 2 and 3 are also difficult to interpret. Why not just use frequency bars? As a whole, several tables and figures are clunky and poorly formatted. This manuscript contains a lot of important information for the reader to digest, and current presentation of this information makes that task unnecessarily difficult and burdensome.
  • The authors use the term “well-being” and similar others (“mental health” and “psychological well-being”) in describing results from the GAD-7. The GAD-7 measures anxiety symptoms, not well-being. The term “well-being” is misleading and should not be used.

Discussion

  • One third of participants were dissatisfied with communication. What practical suggestions do the authors have for improving communication? One of the biggest contributions of this paper is the detailed assessment of what worked and what didn't work with testing rollout. To maximize these data, it would be useful to provide concrete suggestions that others can quickly apply to similar testing programs or environments (e.g., college campuses).
  • Approximately 1 out of every 5 students (~20%) was not confident in their testing results. I was surprised by high this was and also surprised this point was not discussed in the discussion. Why did students lack confidence? For example, was it the timing of the tests, when we (globally) had less information testing? Type of tests? Trust in scientific information and/or medical personnel? From a public health perspective, confidence in testing is essential to uptake. As we continue testing and head into COVID-19 vaccine rollout, there will inevitably be similar public health issues. Thus, this discussion would likely be of value of readers.

Author Response

Thank you for taking the time to review this manuscript. Please find attached our responses.

Reviewer 2 Report

I appreciate the opportunity to review this manuscript titled “Perceptions and experiences of the University of Nottingham pilot asymptomatic testing service: a mixed-methods study”.

I think this is an interesting topic that deserves to be investigated; studying the views of students and staff towards testing programs related to COVID-19 as well as perceptions of the risk of the disease can serve as models to be implemented in other universities. However, I do detect some concerns that should be resolved by the authors.

My first concern is that the authors point to staff as an important part of the study, however, the results they present did not provide much information in relation to this group of participants. Furthermore, a mixed method of study is indicated, however, although the authors present qualitative results, these results were not discussed in depth. In my opinion, you could consider eliminating these results or expanding the discussion.

In the introduction it might be interesting to point out some studies that have been carried out with university students and that show important changes due to the effects of the pandemic. Please include the following manuscript: Meléndez, J. C., Satorres, E., Reyes-Olmedo, M., Delhom, I., Real, E., & Lora, Y. (2020). Emotion recognition changes in a confinement situation due to COVID-19. Journal of Environmental Psychology, 72, 101518.

Regarding the method, I would like to include some elements that could improve your manuscript. You present the P-ATS as a central element of your work. However, the explanation of this program is in supplement 1. You could integrate the current first paragraph of the method with the objectives, since they offer similar information, and start the section with the inclusion of the explanation of the P-ATS that offered in supplement 1. This change would make it easier for the reader to understand without having to go to the supplement.

In addition, you should include the information from Supplement 3 in section 2.1. participants. The authors offer information in the results section that is necessary to know the sociodemographic description of the participants; e.g. of the 99 students who returned the P-AST post, how many were in the first year and how many in the second? When you read the manuscript, you should wait for the results to know this information, and it could be interesting to know it from the description of the participants.

In relation to the GAD-7 scale, there are several concerns that the authors must resolve. The authors could offer more information about the scale, the cut-off point for the diagnosis of generalized anxiety disorder, as well as the reliability obtained in their sample. It would also be important to know what the score range is and its meaning in order to understand the sign of the regressions; When the scale has a negative sign in the regression, what does it mean?

Finally, you talk about mental health (table 2 and other parts of the text). You are applying an anxiety scale and therefore must report the exact construct. Anxiety is not the same as mental health.

In the results section, when the authors present the tables of the regressions, it would be interesting to know the meaning of the signs in the beta; when gender is negative what does it mean; This same information for other variables could facilitate the results to be understood more clearly (e.g. satisfaction).

Author Response

(The authors gave the same response as above.)
